# Prolonged Impella 5.5 Support in Patients with Cardiogenic Shock: A Single-Center Retrospective Analysis

**DOI:** 10.3390/jcm14165631

**Published:** 2025-08-08

**Authors:** Ioana Dumitru, Jonathan DeWolf, Maria Sevillano, LeeAndra Schnell, Hiram Bezerra, Debbie Rinde-Hoffman

**Affiliations:** 1Division of Advanced Heart Failure and Cardiac Transplant, Tampa General Hospital, University of South Florida, Tampa, FL 33606, USA; mariaesevillano@usftgp.org (M.S.); leeandraschnell@usftgp.org (L.S.); hbezerra@usf.edu (H.B.); drhoffman@tgmg.org (D.R.-H.); 2Alabama College of Osteopathic Medicine, Dothan, AL 36303, USA; dewolfj7392@acom.edu

**Keywords:** Impella, cardiogenic shock, heart failure, mechanical circulatory support

## Abstract

**Background:** Patients in cardiogenic shock (CS) often require prolonged mechanical circulatory support lasting longer than 14 days. Prolonged support with Impella 5.5 can improve outcomes in these patients. Here, we describe our experience with prolonged Impella 5.5 support. **Methods:** From January 2023 to June 2024, 64 patients receiving Impella 5.5 support for more than 14 days were identified. Information including demographics, heart failure etiology, and hospital course were collected. **Results:** Of the 64 patients identified, 54 were male, with an average age of 55.9 years. One patient was classified as SCAI class C, 41 as SCAI class D, and 22 as SCAI class E. Heart failure etiologies included 41 patients with non-ischemic cardiomyopathy, 10 with ischemic cardiomyopathy, 10 with acute myocardial infarction, 1 with cardiac allograft rejection, 1 with post-cavotricuspid isthmus ablation, and 1 with post-cardiotomy (aortic root replacement) CS. The average duration of Impella 5.5 support was 27.1 days. Escalation to Impella 5.5 was observed in 26 patients, with 15 having escalated from intra-aortic balloon pump and 11 from Impella CP. Overall survival, including heart recovery, orthotopic heart transplantation (OHT), or left ventricular assist device (LVAD), was 81.3% (52/64) in patients with Impella 5.5. Patients were discharged after OHT (27/64), cardiac recovery (13/64), or durable LVAD placement (12/64), and 12 patients expired. **Conclusions:** Our data suggest that Impella 5.5 provides durable support for patients beyond the 14-day period currently approved by the FDA for CS management. Further evaluation of long-term Impella 5.5 support for cardiac recovery or as a bridge to advanced therapies should be considered.

## 1. Introduction

With the global population aging, the number of people experiencing heart failure (HF) is expected to increase in prevalence by approximately 46% by the year 2030 [1,2]. Despite significant advances in technology, the mortality rate of patients with HF who develop cardiogenic shock (CS) has remained between 30 and 50% [3,4]. Patients with advanced HF in CS often require temporary or long-term mechanical circulatory support (MCS) depending on the underlying conditions of the patient. Several studies have demonstrated that left ventricle (LV) support with MCS can increase the survival rates of patients with CS compared to medical treatment alone [2,5].

Patients with advanced HF in CS who are unresponsive to pharmacologic treatment receive a temporary MCS (tMCS) device to assist with LV unloading as a bridge to heart transplant, durable long-term LV assist device (LVAD) implantation, or cardiac recovery. Patients with CS often present with end organ dysfunction related to reduced cardiac output, such as renal, neurologic, and hepatic dysfunction. This can be mitigated by restoring adequate blood supply with a temporary percutaneous LVAD, helping stabilize the patient’s condition and reverse end organ dysfunction. Patients undergoing tMCS typically follow one of three possible clinical trajectories [6]. First, successful weaning from tMCS for myocardial recovery. Second, tMCS serves as a bridge to decision, providing time for stabilization in patients who are too critically ill for surgery and may require other advanced interventions such as durable LVAD or heart transplant. Lastly, some patients incur irreversible derangements of shock, organ damage, and/or death.

The use of tMCS as a bridge to heart transplant has increased in the United States since the implementation of the new transplant allocation system in 2018, which allows stratification of patients by risk of waitlist mortality [7,8]. Commonly used tMCS devices, including the intra-aortic balloon pump (IABP), venoarterial extracorporeal membrane oxygenation (VA-ECMO), and micro-axial flow pumps such as the Impella, provide varying degrees of hemodynamic support depending on the clinical scenario [9]. As such, standardized guidelines for device selection in CS are lacking, and decisions are often driven by institutional protocols and local clinical expertise. IABP provides modest circulatory support and is often used in cases of mild CS or high-risk percutaneous coronary intervention, while VA-ECMO delivers full cardiopulmonary support and is typically reserved for patients with severe biventricular failure or cardiac arrest [10]. The benefits of Impella (Abiomed, Danvers, MA, USA) tMCS support over these other devices, however, include LV unloading and decreased oxidative stress on the heart, which facilitate better overall recovery while allowing patient mobilization and rehabilitation. Moreover, unloading of the LV improves hemodynamics, including pulmonary arterial pressure (PAP), right atrial pressure (RAP), pulmonary capillary wedge pressure (PCWP), and, indirectly, right ventricle (RV) function. In addition, Impella support provides improved organ perfusion, thereby mitigating the risk of end organ dysfunction.

The Impella 5.5 percutaneous micro-axial pump provides increased flow rates (up to 5.5 L/min) compared to previous Impella devices (Impella CP, 5.0). In addition, Impella 5.5 is shorter than the 5.0, more rigid, and lacks a pig tail, which may reduce the incidence of thrombosis [11]. Moreover, with short-term use, Impella 5.5 has been shown to increase the weaning rate in patients with acute myocardial infarction with CS (AMICS), cardiomyopathy, and post-cardiotomy CS (PCCS) [11]. Lastly, the implant procedure for the Impella 5.5 device is minimally invasive via the axillary artery, which allows for better patient mobility and mitigates the deleterious effects of bed rest on the cardiovascular, musculoskeletal, and central nervous systems [12].

Overall, the use of tMCS devices as a bridge to transplant or durable LVAD has increased, as they provide an effective and less invasive approach to stabilize patients with CS while they wait for their procedures. In some cases, the need for tMCS extends beyond the short-term FDA-approved 14 days for Impella 5.5. Multiple studies have already shown that extended use of Impella 5.5 is safe and effective with a low incidence of mechanical failure [13,14,15,16,17,18,19,20,21,22,23,24,25,26,27]. Here, we describe our experience with prolonged Impella 5.5 support lasting more than 14 days.

## 2. Methods

### 2.1. Study Population

This was a single-center retrospective analysis which identified a total of 64 patients with CS from January 2023 to June 2024 who received tMCS support with Impella 5.5 for ≥14 days. Patients were treated at Tampa General Hospital, a quaternary care hospital with a dedicated HF and CS team. The retrospective nature of this study precluded the requirement for informed patient consent. Patient demographics including HF etiology and SCAI stage at admission are presented in Table 1.

### 2.2. Data Collection

Patient hemodynamics, laboratory, and inotropic support were compared pre- and post-Impella 5.5 implant. Patient inotropic support was divided into three groups: low-dose (dobutamine ≤ 5, or milrinone < 0.5 mg/kg/min), high-dose (dobutamine > 5, or milrinone > 0.5 mg/kg/min), and no inotropic support (Table 2). Patient vasopressor (vasopressin/norepinephrine) and anticoagulant therapy were also quantified and presented in Table 2. Key laboratory metrics used to assess cardiac, hepatic, and renal dysfunction included lactate, alanine transaminase (ALT), bilirubin, and creatinine. The hemodynamic parameters evaluated included right atrial pressure (RAP), pulmonary artery pressure (PAP; systolic and diastolic), pulmonary capillary wedge pressure (PCWP), pulmonary artery saturation (PAS), and tricuspid annular plane systolic excursion (TAPSE). Key laboratory and hemodynamic metrics are presented in Table 3.

### 2.3. Outcomes

The primary outcomes of this study were duration of Impella 5.5 support, escalation to Impella 5.5 from IABP or Impella CP, and the need for escalation to ECMO or biventricular CentriMag (Abbott, Chicago, IL, USA) devices. Final patient disposition outcomes were recorded as discharge, cardiac recovery, heart transplant, placement of a durable LVAD, or death. The number of patients requiring inpatient hemodialysis and those who were readmitted within 30 days for an HF condition were also tabulated. All data are presented as mean (minimum, maximum) unless otherwise indicated.

## 3. Results

### 3.1. Patient Demographics

Over the course of this study, 64 patients received Impella 5.5 for CS and required tMCS support for longer than the current FDA-approved duration of 14 days. Of these patients, 54/64 (84.4%) were male with an average age of 55.9 (18, 78) years and an average body mass index of 28.2 (18.2, 51.4) kg/m^2^. Most patients had cardiomyopathy, with 64.1% (41/64) having non-ischemic cardiomyopathy and 15.6% (10/64) having ischemic cardiomyopathy. Ten patients (15.6%) experienced an acute myocardial infarction. Of the three remaining patients, one was post-cardiotomy, one developed CS following cavotricuspid isthmus ablation, and one developed HF due to cardiac allograft rejection. As a cohort, 1.6% (1/64) were classified as SCAI Stage C, 64.1% (41/64) as SCAI Stage D, and 34.4% (22/64) as SCAI Stage E.

### 3.2. Pharmacologic Support

Prior to Impella 5.5 placement, 56.3% (36/64) of patients received low-dose (dobutamine ≤ 5, or milrinone ≤ 0.5 mg/kg/min) inotrope support, while 32.8% (21/64) received high-dose (dobutamine > 5, or milrinone > 0.5 mg/kg/min) inotrope support (Table 2). The remaining 10.9% of patients (7/64) did not require inotrope support. A total of 73.4% (47/64) of patients also received vasopressors (norepinephrine or vasopressin) and 95.3% (61/64) of patients were administered systemic heparin. Additionally, bicarbonate was used as the Impella purge solution in 84.4% (54/64) of patients, heparin was used in 14.1% (9/64), and a combination of both was used in 3.1% (2/64) of patients. One patient was given protamine for an axillary hematoma, and two patients were administered thrombolytic therapy due to a cerebrovascular accident.

### 3.3. Laboratory and Hemodynamics

Key laboratory and hemodynamic metrics are summarized in Table 3. Of note, not all patient charts examined contained complete laboratory and hemodynamic records. Therefore, the values presented here reflect those of patients with complete pre- and post-Impella 5.5 placement records for either laboratory (*n* = 19) or hemodynamic (*n* = 10) measures. The mean lactate level pre-Impella 5.5 placement was 2.1 (1.0, 1.9) µmol/L, which decreased to 1.18 (0.5, 1.4) µmol/L post-Impella 5.5 placement. The mean ALT was 791.4 (18.5, 178.0) U/L pre-Impella 5.5 placement, which decreased to 30.8 (14.3, 28.0) U/L post-Impella 5.5 placement. Three patients in this group had ALT values > 1700 U/L, with one patient having a level of 10,396 U/L, indicating acute hepatic dysfunction likely attributed to ischemic injury. Following Impella 5.5 implantation, this patient’s ALT level dropped to 42 U/L, indicating hepatic recovery, and the patient was subsequently discharged. Mean bilirubin levels were 2.4 (0.8, 3.5) μmol/L pre-Impella 5.5 placement and decreased to 1.7 (0.7, 2.2) μmol/L post-Impella 5.5 placement. Serum creatinine had a mean level of 2.1 (1.1, 3.0) mg/dL pre-Impella 5.5 placement and 1.5 (1.0, 1.6) mg/dL post-Impella 5.5 placement.

In the subgroup of patients with complete hemodynamic records, the pre-Impella 5.5 placement RAP was 15.6 (9.3, 19.8) mmHg, which fell to 11.8 (9.0, 14.0) mmHg following Impella 5.5 placement. Pre-Impella 5.5 pulmonary systolic and diastolic pressures were 49 (38.8, 57.3) mmHg and 26.4 (16.3, 40.0) mmHg, which were relatively maintained post-Impella 5.5 placement at 48 (29.0, 67.0) mmHg and 22.5 (17.3, 30.0) mmHg, respectively. Moreover, PCWP was 24.2 (17.8, 28.5) mmHg pre-Impella 5.5 placement and 21.8 (16.5, 28.0) mmHg post-Impella 5.5 placement. Pulmonary arterial O_2_ saturation levels were 59.5 (46.0, 74.8) mmHg and 60.6 (50.0, 73.0) mmHg pre- and post-Impella 5.5 placement, respectively. Finally, TAPSE was 1.18 (1.0, 1.4) cm pre-Impella 5.5 placement and 1.9 (1.5, 2.4) cm post-Impella 5.5 placement.

### 3.4. Patient Outcomes

Among the 64 patients identified in this study, the average duration of Impella 5.5 support was 27.1 (14, 76) days. Escalation to Impella 5.5 was observed in 40.6% (26/64) of patients, 57.7% (15/26) from IABP, and 42.3% (11/26) from Impella CP. Escalation beyond Impella 5.5 was observed in 12.5% (8/64) of patients, of which 87.5% (7/8) escalated to ECPella (the combination of Impella and ECMO), and 12.5% (1/8) escalated to biventricular CentriMag support. Overall recovery to discharge, including heart recovery, orthotopic heart transplantation (OHT), or durable LVAD, was 81.3% (52/64). Of the patients discharged, 51.9% (27/52) were discharged after an OHT, 25% (13/52) after cardiac recovery, and 23.1% (12/52) after durable LVAD placement. Study outcomes are summarized in Table 4.

### 3.5. Complications

The most prevalent complications within this cohort included hemolysis, defined by lactose dehydrogenase levels > 550 U/L, occurring in 67.2% (43/64) of patients, arrhythmias in 65.6% (42/64), bleeding requiring blood transfusion during Impella 5.5 support in 60.9% (39/64), and acute kidney injury (AKI), defined by elevated creatinine levels, in 53.1% (34/64) of patients, with 17.2% (11/64) requiring inpatient hemodialysis. Vascular injury occurred in 18.8% (12/64) of patients, and hematomas were noted in 45.3% (29/64). There were two instances of Impella 5.5 pump dysfunction due to the development of thrombosis within the device. Patient complications are summarized in Table 5.

## 4. Discussion

In the current study, we demonstrate that in patients with CS who require tMCS, implementation of Impella 5.5 support for longer than the current FDA-approved 14-day duration is a safe and effective support strategy at our institution. For the 64 patients included in this study, Impella 5.5 support lasted between 14 and 76 days, with an overall survival rate of 81.3% (52/64), which aligns with other reported studies on long-term Impella 5.5 use [17,20,28]. Moreover, the mortality rate for long-term Impella 5.5 support in this cohort was 18.7% (12/64), which is a substantial improvement over the established long-term mortality rate of 30–50% observed in patients with medically-treated CS.

In patients who develop CS, early identification and rapid intervention, including tMCS deployment, are crucial to achieving improved survival, reducing morbidity, and promoting cardiac and end organ recovery [29]. There is currently a paucity of prospective clinical trial data evaluating the implementation of tMCS interventions in CS, as studying CS interventions in a randomized fashion is extremely challenging given the confounding complexity of etiologies, associated comorbidities, and the variable timeframe in which CS evolves. However, the National Cardiogenic Shock Initiative and the Danish–German CS randomized trial both demonstrated that early LV unloading with Impella MCS devices leads to improved survival when deployed in patients with CS, supporting the value of their use in improving patient outcomes [30,31].

Impella devices have several potential advantages over other tMCS modalities, such as ECMO or IABP. While IABP can enhance coronary perfusion, it does not provide the same level of direct cardiac support as Impella, which completely unloads the LV, thus reducing cardiac workload and improving overall circulatory dynamics [32]. ECMO offers comprehensive circulatory support and is indicated when a patient with CS requires pulmonary support, but is more invasive than Impella, does not unload the LV, and is associated with higher complication rates [33]. Furthermore, Impella devices, particularly the axillary-implanted Impella 5.5, facilitate patient mobility due to their design, preventing complications associated with prolonged bed rest, such as pressure ulcers and muscle deconditioning. Indeed, mobilization during Impella support has been associated with better functional outcomes [34].

The axillary position of the Impella 5.5 is also quite stable, minimizing dislocation and hemolysis complications and allowing the device to remain in place for a longer duration as a bridge to durable LVAD or heart transplant [35]. While the current FDA-approved use duration for the Impella 5.5 is 14 days, studies evaluating its performance have found the pump to be durable, displaying a very low incidence of pump malfunction when used for treatment durations extending out to 123 days [13,14,15,16,17,18,19,20,21,22,23,24,25,26,27]. In the cases analyzed during this study, only one Impella device developed a mechanical problem due to thrombus formation that required the pump to be replaced. A pump thrombosis was discovered in one additional patient, which was managed with medical treatment without requiring pump replacement. Additionally, the prolonged duration has not been found to have an adverse impact on survival rates, with one study of 332 heart transplant patients demonstrating comparable 90-day survival rates between patients receiving support for <14 days and those supported for >14 days [20]. The survival rate from larger studies with Impella 5.5 support ranged from ~67 to 95% [13,15,17,20,25,28], reinforcing the assertion that Impella 5.5 is safe and effective for long-duration MCS.

In this study, the most common complications observed were AKI, arrhythmia, bleeding, and hemolysis. The hemolysis rate in this cohort of patients was higher than what has been observed in other studies [11,36], in which it ranged from ~3% to 43%. Given the multifactorial contributions to hemolysis, including the severity of illness, anticoagulation protocols, and device positioning, our protocolized approach aims to minimize rates of hemolysis through the use of echocardiography with contrast to assess LV thrombus, repositioning in response to alarms, decreasing Impella support level, and increasing anticoagulation. While the percentage of patients experiencing AKI was slightly more than half in this cohort, only a fraction of these required inpatient hemodialysis, and most recovered prior to discharge. It is challenging to attribute these complications to the use of Impella, as the patients were critically ill, mainly SCAI stage D and E, with high morbidity and mortality, end organ dysfunction due to CS, and exposure to multiple invasive procedures outside of Impella placement and full anticoagulation.

## 5. Study Limitations

This study is limited by its retrospective nature and includes only a limited number of patients. Additionally, this study lacked a control group of patients with similar clinical characteristics who were not treated with Impella, limiting our ability to directly compare outcomes across treatment strategies. Since complete laboratory and hemodynamic values were not available for all patients, the statistical power for this patient population is insufficient to provide adequate interpretations for future clinical implications. As such, larger prospective and comparative studies are needed to fully evaluate the effectiveness of long-term Impella 5.5 use in patients with CS.

## 6. Conclusions

Currently, Impella 5.5 is FDA-approved for less than 14 days of CS management. However, data from our center suggests that the pump has the durability to support patients successfully for up to one month and possibly beyond. While this is a small dataset, further evaluation of long-term Impella 5.5 support for cardiac recovery and/or as a bridge to advanced therapies is needed. These findings add to the growing body of evidence suggesting that Impella 5.5 is a promising intermediate percutaneous LV tMCS with durability beyond the FDA-approved use duration, providing critical prolonged support in stabilizing patients with CS until they are candidates for further interventions such as durable LVAD or heart transplant.

## Figures and Tables

**Table 1 jcm-14-05631-t001:** Patient demographics.

Characteristic	*n* = 64
Age, years, (min, max)	55.9 (18, 78)
Sex, male, *n* (%)	54/64 (84.4)
BMI, kg/m^2^, (min, max)	28.2 (18.2, 51.4)
Etiology of HF, *n* (%)	
NISCMP	41/64 (64.1)
ISCMP	10/64 (15.6)
Acute MI	10/64 (15.6)
Allograft rejection	1/64 (1.56)
Post-CTI ablation	1/64 (1.56)
Post-cardiotomy ^a^	1/64 (1.56)
SCAI stage at admission, *n* (%)	
C	1/64 (1.6)
D	41/64 (64.1)
E	22/64 (34.4)

^a^ aortic root replacement, cardiogenic shock. Abbreviations: BMI, body mass index; CTI, cavotricuspid isthmus; HF, heart failure; ISCMP, ischemic cardiomyopathy; max, maximum value; min, minimum value; MI, myocardial infarction; NISCMP, non-ischemic cardiomyopathy; SCAI, Society for Cardiovascular Angiography and Intervention.

**Table 2 jcm-14-05631-t002:** Pharmacologic support.

Characteristic	*n* = 64
Short-term pharmacology, *n* (%)	
Inotropes	
None	7/64 (10.9)
Low-dose ^a^	36/64 (56.3)
High-dose ^b^	21/64 (32.8)
Pressors ^c^	47/64 (73.4)
Anticoagulant therapy, *n* (%)	
Systemic anticoagulant	
Heparin	61/64 (95.3)
Non-heparin ^d^	3/64
Heparin purge	9/64 (14.1)
Sodium bicarbonate purge	54/64 (84.4)
Heparin and sodium bicarbonate purge	2/64 (3.1)

^a^ Low-dose: dobutamine ≤ 5, milrinone ≤ 0.5 mg/kg/min; ^b^ high-dose: dobutamine > 5, milrinone > 0.5 mg/kg/min; ^c^ pressors: vasopressin or norepinephrine; ^d^ bivalirudin or argatroban.

**Table 3 jcm-14-05631-t003:** Laboratory and hemodynamic values.

Parameter, Mean (min, max)	*Pre-Impella ^a^*	*Post-Impella ^a^*
Key laboratory parameters (*n* = 19)		
Lactate, mmol/L	2.1 (1.0, 1.9)	1.18 (0.5, 1.4)
ALT, U/L	791.4 (18.5, 178.0)	30.8 (14.3, 28.0)
Bilirubin, μmol/L	2.4 (0.8, 3.5)	1.7 (0.7, 2.2)
Creatinine, mg/dL	2.1 (1.1, 3.0)	1.5 (1.0, 1.6)
Key hemodynamic parameters (*n* = 10)		
RA pressure, mmHg	15.6 (9.3, 19.8)	11.8 (9.0, 14.0)
PA systolic, mmHg	49 (38.8, 57.3)	48 (29.0, 67.0)
PA diastolic, mmHg	26.4 (16.3, 40.0)	22.5 (17.3, 30.0)
PCWP, mmHg	24.2 (17.8, 28.5)	21.8 (16.5, 28.0)
PA Sat, mmHg	59.5 (46.0, 74.8)	60.6 (50.0, 73.0)
TAPSE, cm	1.2 (1.0, 1.4)	1.9 (1.5, 2.4)

^a^ Mean values were calculated only from patients with complete laboratory and hemodynamic records for each category, as some patients did not have complete pre- and post-Impella data available. Abbreviations: ALT, alanine transaminase; max, maximum value; min, minimum value; PA, pulmonary artery; PCWP, pulmonary capillary wedge pressure; RA, right atrial; Sat, saturation; TAPSE, tricuspid annular plane systolic excursion.

**Table 4 jcm-14-05631-t004:** Patient outcomes.

Outcomes	*n* = 64
Duration of Impella 5.5 support (days), mean (min, max)	27.1 (14, 76)
Escalation to Impella 5.5, *n* (%)	26/64 (40.6)
From IABP	15/64 (23.4)
From Impella CP	11/64 (17.2)
Escalation beyond Impella 5.5, *n* (%)	
ECPella	7/64 (10.9)
Biventricular CentriMag	1/64 (1.6)
Patient disposition, *n* (%)	
Recovered to discharge	52/64 (81.3)
Cardiac recovery	13/64 (20.3)
Received OHT	27/64 (42.2)
Received LVAD	12/64 (18.2)
Expired *	12/64 (18.7)
30-Day HF readmission, *n* (%)	8/52 (15.4)

* Includes one patient transferred to hospice. Abbreviations: HF, heart failure; IABP, intra-aortic balloon pump; LVAD, left ventricular assist device; max, maximum value; min, minimum value; OHT, orthotopic heart transplantation.

**Table 5 jcm-14-05631-t005:** Impella complications.

Complication (*n* = 64)	Incidence, *n* (%)
AKI	34/64 (53.1)
AKI Requiring HD	11/64 (17.2)
Arrhythmia ^a^	42/64 (65.6)
Bleeding ^b^	39/64 (60.9)
CVA	6/64 (9.4)
Hematoma ^c^	29/64 (45.3)
Hemolysis ^d^	43/64 (67.2)
Infection ^e^	25/64 (39.1)
Pump Stoppage	1/64 (1.6)
Thrombosis	3/64 (4.7)
Vascular Injury ^f^	12/64 (18.8)
Inpatient Hemodialysis	19/64 (29.7)
None	1/64 (1.6)

^a^ Arrhythmia: requiring ICD shocks, DCCV, or antiarrhythmic medications; ^b^ Bleeding: requiring blood product transfusions during Impella support; ^c^ Hematoma: localized bleeding outside of blood vessels; ^d^ Hemolysis: identified by lactate dehydrogenase (LDH) > 550 U/L (which is twice the upper value of normal); ^e^ Infection: infection not related to Impella, required systemic antibiotics; ^f^ Vascular injury: requiring vascular repair, or as evident for other venous complications. Abbreviations: AKI, acute kidney injury; CVA, cerebrovascular accident; DCCV, direct current cardioversion; HD, hemodialysis; ICD, implantable cardioverter-defibrillator.

## Data Availability

The original contributions presented in this study are included in the article. Further inquiries can be directed to the corresponding author.

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
