# Peer review of "Prolonged Impella 5.5 Support in Patients with Cardiogenic Shock: A Single-Center Retrospective Analysis"

_jcm, 2025, doi:10.3390/jcm14165631_

Round 1

Reviewer 1 Report

Comments and Suggestions for Authors

This is a timely and well-written retrospective study describing the use of prolonged Impella

5.5 support (>14 days) in patients with cardiogenic shock (CS). The authors present

real-world data from a single tertiary center, including patient characteristics, hemodynamic

and laboratory data, and clinical outcomes. The study adds valuable insight into the potential

role of Impella 5.5 as a bridge to recovery or advanced therapies beyond the FDA-approved

duration.

The manuscript is generally clear and well-organized. However, there are some areas that

require clarification and elaboration to strengthen the scientific rigor and clarity of the paper.

Justification of the Study

The introduction could better highlight the existing knowledge gap. Specifically, why is

studying support beyond 14 days clinically relevant? What novel information does this cohort

contribute compared to previous studies

Patient Selection

The manuscript should describe the inclusion and exclusion criteria more explicitly to clarify

how patients were selected for prolonged support.

Complications

The observed hemolysis rate (67.2%) is notably higher than reported in previous studies.

The authors could discuss potential contributing factors, such as patient severity,

anticoagulation protocols, or device positioning.

Figures and Tables

Consider adding visual representation to enhance readability.

Revised Title (suggestion)

Prolonged Impella 5.5 Support in Patients With Cardiogenic Shock: A Single-Center

Retrospective Analysis

Author Response

Reviewer #1

Comment 1: The introduction could better highlight the existing knowledge gap. Specifically, why is studying support beyond 14 days clinically relevant? What novel information does this cohort contribute compared to previous studies

Response 1:  We appreciate the reviewer’s comment and have added language to the introduction to better highlight the existing knowledge gap. While the Impella 5.5 is approved for short-term use (typically up to 14 days), in clinical practice, patients with severe cardiogenic shock (CS) often require hemodynamic support beyond this period. However, there is limited data on the safety, durability, and outcomes associated with prolonged use. Understanding the risks of complications—such as hemolysis, infection, thrombosis, and device malfunction—as well as the potential benefits of extended support is crucial for guiding clinical decision-making. Our study begins to address this gap by examining real-world outcomes in patients who received Impella 5.5 support for more than 14 days.

Comment 2: The manuscript should describe the inclusion and exclusion criteria more explicitly to clarify how patients were selected for prolonged support.

Response 2: As this was a retrospective study, no additional predefined inclusion or exclusion criteria were applied beyond identifying patients with cardiogenic shock (CS) who received Impella 5.5 support for more than 14 days. All eligible cases meeting this criterion during the study period were included. The decision to extend support beyond 14 days was made by the treating clinical teams based on individual patient needs. This study aimed to evaluate the outcomes associated with such prolonged use in real-world clinical practice.

Comment 3: The observed hemolysis rate (67.2%) is notably higher than reported in previous studies. The authors could discuss potential contributing factors, such as patient severity, anticoagulation protocols, or device positioning.

Response 3: We have added this to the Discussion on page 11.

Comment 4: Consider adding visual representation to enhance readability.

Response 4: Thank you for the helpful suggestion. While we agree that visual representations can enhance readability in some cases, we believe that the data in this study are most clearly and appropriately presented in tabular format. Additionally, as graphical abstracts are not required by the journal, we respectfully decline to add a visual representation at this time. We hope the current presentation sufficiently conveys the key findings.

Comment 5: Suggest revised title “Prolonged Impella 5.5 Support in Patients With Cardiogenic Shock: A Single-Center Retrospective Analysis”

Response 5: We thank the reviewer for this suggestion and have updated the manuscript title accordingly.

Reviewer 2 Report

Comments and Suggestions for Authors

The authors of the manuscript share their clinical experience  from treating 64 patients with cardiogenic shock who have recieved Impella 5.5 device support for more than 14 days. The manuscript is important from clinical point of view. Cardiogenic shock is an acute life-threatening event with high mortality rates. Nowadays, it is due mostly to acute anterior myocardial infarction, ventricular tachycardia/flutter/fibrillation, end-stage chronic heart failure or drugs/factors supressing significantly myocardial contractility. Device treatment in some of these patients is crucial for their immediate and long-term prognosis and Impella devices are among the most preferred ones. 

I have the following comments and recommendations about this manuscript:

1. The "Introduction" must be shortened and focused on device treatment options in cardiogenic shock.

2. Clear inclusion and exclusion criteria for this study must be formulated.

3. The type of the study design must be specified in the beginning of "Methods"

4. A major weakness of this study is the lack of a control group with similar characteristics but treated without Impella to compare the clinical outcomes between the groups. 

5. P values are missing in the tables (particlularly important for Table 3 - comparison between pre-Impella and post-Impella parameters).

Author Response

Reviewer #2:

Comment 1: The "Introduction" must be shortened and focused on device treatment options in cardiogenic shock.

Response 1: We have carefully considered both comments and believe that the current version strikes an appropriate balance by providing essential context on device treatment options for cardiogenic shock, while also clearly highlighting the knowledge gap that our study aims to address. Accordingly, we have chosen to retain the current structure and content, as we feel it effectively supports the study rationale and aligns with the manuscript’s objectives. To further strengthen this section, we have added additional details on the specific devices used in the treatment of cardiogenic shock and the current challenges surrounding device selection in the Introduction on pages 3 and 4.

Comment 2: Clear inclusion and exclusion criteria for this study must be formulated.

Response 2: This was a retrospective study, with no additional predefined inclusion or exclusion criteria applied beyond identifying patients with cardiogenic shock (CS) who received Impella 5.5 support for more than 14 days. All eligible cases meeting this criterion during the study period were included. The decision to extend support beyond 14 days was made by the treating clinical teams based on individual patient needs in all cases. This study aimed to evaluate the outcomes associated with such prolonged use in real-world clinical practice.

Comment 3: The type of the study design must be specified in the beginning of "Methods"

Response 3: As requested, we have updated the beginning of the Methods section on page 5, which now specifies that this study was a single-center retrospective analysis.

Comment 4: A major weakness of this study is the lack of a control group with similar characteristics but treated without Impella to compare the clinical outcomes between the groups. 

Response 4: We acknowledge the lack of a control group as a limitation and have added this point to the limitations section of the manuscript on page 11. Due to the retrospective nature of the study and the severity of illness in this cohort, identifying a comparable group was not feasible. Nonetheless, the study provides important real-world data on extended Impella support, where evidence remains limited. Future comparative studies are needed to better assess relative outcomes across support strategies.

Comment 5: P values are missing in the tables (particularly important for Table 3 - comparison between pre-Impella and post-Impella parameters).

Response 5: We appreciate the reviewer’s comment regarding the inclusion of P values in Table 3. However, for parameters in Table 3, we have opted not to include formal statistical comparisons, as these values represent a small subset of the cohort. Given the retrospective nature of the study and incomplete data availability, statistical tests for Table 3 would not be representative of the entire population and may lead to overinterpretation as stated in the limitations section of the manuscript on page 11.

Editorial comments:

Comment 1 and 2: We noticed that the main text of your manuscript is quite brief which may mean that the methods, discussion, or future research directions are not described in enough detail. Please consider the following points in your revisions: adding full inclusion/exclusion details and expanding the discussion section.

Response 1 and 2: We acknowledge that the manuscript is relatively concise; however, we have carefully ensured that all relevant aspects of the study—including the methods, key findings, and implications for future research—are appropriately addressed. We have aimed to maintain clarity and focus without adding unnecessary length, and we believe the current level of detail sufficiently supports the study’s objectives and conclusions.

Round 2

Reviewer 2 Report

Comments and Suggestions for Authors

The authors have addressed most of the reviewer's recommendations and have made the corresponding correcions/additions to their manuscript. The manuscript hs been significantly improved and is now ready to be published in its current version.

Author Response

Comment 1: The authors have addressed most of the reviewer's recommendations and have made the corresponding correcions/additions to their manuscript. The manuscript hs been significantly improved and is now ready to be published in its current version.

Response 1: We thank the reviewer for their encouraging words and appreciate the time taken for this peer-review.